# Knowledge and Attitude Regarding Monkeypox Virus among Physicians in Saudi Arabia: A Cross-Sectional Study

**DOI:** 10.3390/vaccines10122099

**Published:** 2022-12-08

**Authors:** Najim Z. Alshahrani, Mohammed R. Algethami, Abdullah M. Alarifi, Faris Alzahrani, Eman A. Alshehri, Aishah M. Alshehri, Haytham Abdulwhab Sheerah, Abdelaziz Abdelaal, Ranjit Sah, Alfonso J. Rodriguez-Morales

**Affiliations:** 1Department of Family and Community Medicine, Faculty of Medicine, University of Jeddah, Jeddah 21589, Saudi Arabia; 2Ministry of Health, Jeddah 21589, Saudi Arabia; 3Department of Public Health, College of Health Sciences, Saudi Electronic University, Riyadh 13323, Saudi Arabia; 4Department of Public Health, General Directorate of Health Affairs in Aseer Region, Ministry of Health, Abha 62529, Saudi Arabia; 5General Directorate of Health Affairs in Aseer Region, Ministry of Health, Abha 62529, Saudi Arabia; 6Harvard Medical School, Boston, MA 02115, USA; 7Institute of Medicine, Tribhuvan University Teaching Hospital, Kathmandu 44600, Nepal; 8D.Y Patil Medical College, Hospital and Research Center, Dr. D.Y. Patil Vidyapeeth, Pune 411018, Maharashtra, India; 9Grupo de Investigación Biomedicina, Faculty of Medicine, Fundación Universitaria Autónoma de Las Américas—Institución Universitaria Vision de Las Americas, Pereira 660003, Risaralda, Colombia; 10Master of Clinical Epidemiology and Biostatistics, Universidad Científica del Sur, Lima 15024, Peru; 11Gilbert and Rose-Marie Chagoury School of Medicine, Lebanese American University, Beirut P.O. Box 36, Lebanon; 12Latin American Network of MOnkeypox VIrus Research (LAMOVI), Pereira 660003, Risaralda, Colombia

**Keywords:** attitude, monkeypox virus, knowledge, physicians, Saudi Arabia

## Abstract

The growing incidence of human monkeypox cases emphasizes the significance of prevention, early detection, and prompt responses for healthcare providers. The aim of this study was to assess the knowledge and attitudes toward monkeypox infection among physicians, a frontline healthcare worker group, in Saudi Arabia. A cross-sectional survey assessing knowledge and attitudes towards monkeypox infection on multiple-item scales was sent to physicians in Saudi Arabia. The associations between independent factors and either knowledge or attitude were assessed. The final analysis included 398 participants. Approximately 55% of the surveyed participants had a “good knowledge” score about human monkeypox. The adjusted logistic regression analysis showed that being a female physician, working in the private sector, and having information on human monkeypox during medical school or residency years were the only factors associated with a good level of knowledge about human monkeypox. However, physicians’ knowledge and attitudes regarding monkeypox infection are inadequate and influenced by various factors. There is a significant knowledge gap between the therapeutic management of monkeypox and its vaccination. Training and knowledge assessments are important, especially when studies show significant improvement in related and specific knowledge.

## 1. Introduction

Human Monkeypox (HMPX), a zoonotic infectious disease caused by the Monkeypox Virus (MPXV) of the genus Orthopoxvirus, causes smallpox-like symptoms in humans [1]. It is a predominantly endemic disease in Western and Central Africa [2]. The name, monkeypox, originates from the first reported discovery of the virus as an outbreak of a pox-like disease in monkeys at an animal research facility in Copenhagen, Denmark [3]. The first HMPX case in medical history was recorded in 1970 at a hospital in the Democratic Republic of Congo, when a nine month-old child manifested smallpox-like symptoms [4]. In 2003, multiple cases of HMPX were reported within the U.S., representing the first confirmed cases of HMPX outside the African continent [5]. Fourteen years later, one of the most significant monkeypox outbreaks was reported in Nigeria, with 197 suspected cases and sixty-eight confirmed cases [5]. In May 2022, the largest outbreak of MPXV outside of endemic regions was confirmed by the World Health Organization (WHO) [6]. The 2022 outbreak marks the first time MPXV spread extensively outside of Western and Central Africa. The continuous outbreak of MPXV has gained widespread attention. First concentrated in Central and West Africa, the novel waves of the monkeypox epidemic have experienced a dramatic amplification since May 2022 [1,7]. As of 1 December 2022, 81,225 cases and 56 deaths have been reported, affecting 110 regions around the world [8]. Many of these cases have been reported to have a vesicular rash illness in men who have had sex with men (MSM) [9].

Monkeypox can be transmitted mainly via contact with respiratory secretions, infected skin lesions, or contaminated materials [10]. The incubation period of monkeypox usually lasts from six to thirteen days but can range from five to twenty-one days [11]. The disease is often self-limiting, with symptoms occurring spontaneously within fourteen to twenty-one days [4,10,12]. Symptoms can range from mild to severe, and lesions can be very itchy or painful. The animal reservoir remains unknown, although it is likely to be among rodents [13]. Contact with live and dead animals through hunting and consumption of wild game or bush meat are known risk factors [14]. The disease’s clinical manifestations are comparable to but less severe than smallpox [12]. Lymphadenopathy seems to be one of the key differentiating factors between monkeypox and smallpox [15]. HMPX symptoms include fever, generalized headache, fatigue, lymphadenopathy, back pain, myalgia, and rash [10,12].

Historically, vaccination against smallpox was shown to be protective against monkeypox, and reported to be 85% [16]. However, while one vaccine (MVA-BN) and one specific treatment (tecovirimat) were approved for monkeypox in 2019 and 2022, respectively, these countermeasures are not yet widely available, and populations worldwide under the age of 40 or 50 years no longer benefit from the protection afforded by prior smallpox vaccination programs [16,17].

The increased number of HMPX cases demonstrates healthcare workers’ importance in terms of the prevention, early detection, and quick response/management of monkeypox. However, a report by the WHO showed that one of the challenges faced in preventing the re-emergence of monkeypox was a lack of knowledge of monkeypox, particularly among healthcare workers. There have been several studies done to assess knowledge, attitude, and practice (KAP) among general population and Healthcare Workers (HCWs). A study done by Sallam et al. [18] found unsatisfactory levels (33.3%) of HMPX knowledge among HCWs. Another study done among Saudi medical students found that the vast majority of them (72%) had poor knowledge about the monkeypox virus [19]. Therefore, healthcare workers must be knowledgeable and well-prepared for monkeypox cases in different regions, including the Middle East. In this region, especially in Saudi Arabia, there are the Hajj and Omrah destinations, which could increase its vulnerability to the importation of HMPX. Hence, we sought to assess physicians’ knowledge and attitudes toward monkeypox in Saudi Arabia.

## 2. Materials and Methods

### 2.1. Study Design and Setting

An analytical cross-sectional study was conducted among physicians in Saudi Arabia from 26 March 2022 to 27 May 2022. This study recruited registered Saudi physicians who practice various medical specialties and work in various Saudi regions. Non-Saudi physicians who work outside of Saudi Arabia and participants who did not provide informed consent were excluded. We calculated the sample size by using the Raosoft calculator. We determined the sample size based on recent data, estimating the total number of physicians in Saudi Arabia to be 114,000 [20]. Since no previous studies in Saudi Arabia examined physicians’ knowledge of monkeypox, a conservative estimate of 50 percent was used. The minimum sample size required for a 95% confidence interval with a margin of error of 5% was 383. The size of the sample was increased because it was thought that fewer people would fill out an online questionnaire.

### 2.2. Data Collection Process

We used the convenience sampling technique to recruit participants; no monetary benefit was offered to any participant. The participants were invited to participate in the questionnaire using Google Template, and they were approached using social media platforms (WhatsApp and Twitter). Before participating, each respondent was asked for their informed consent by clicking on the consent statement before attempting to fill out the responses. The informed consent statement contains the purpose of this study and details of its objectives. Additionally, it stated, “I hereby, after reading the aims of the study, engage in the survey, supplying my information by answering questions intelligently and voluntarily.” Participants completed the survey and clicked the “submit” button to send it to our platform for data gathering. To submit a legitimate response, all of the questions had to be answered.

### 2.3. Data Collection Tools

This study was conducted using a self-administered questionnaire (see Appendix A) adapted based on current information from the Centers for Disease Control and Prevention (CDC) and some previously published studies [16,17,18,19,20,21]. Then, the questionnaire was modified by two preventive medicine consultants to fit the objectives and scope of the current research. Finally, the questionnaire was pre-tested on twenty physicians by completing the questionnaire at two points in time (i.e., 3 March 2022 and 19 March 2022) for a test–retest of its reliability. Items whose corresponding correlation coefficient in T1 vs. T2 was >0.80 were considered “consistent” and were ultimately included in the final questionnaire that was then delivered by 26 March 2022.

The questionnaire was divided into four sections. The first two sections addressed sociodemographic factors such as age, gender, marital status, level of work, medical specialty, and work sector, followed by medical practice experience. The third section covered monkeypox knowledge, which included 23 multiple-choice questions (MCQs) with a score of one for a correct answer and a score of zero for an incorrect answer. When the scores were added up, a total score from 0 to 23 was given. Higher scores showed more knowledge. The fourth section was about physicians’ attitudes towards monkeypox and was assessed by ten statements. These statements were answered on a 3-point Likert scale (the answers were either agree, have no opinion, or disagree). These statements included opinions about the ability of the world’s populations to control the monkeypox epidemic and the presence of suitable preventive and control measures. They were also asked how interested they were in learning more about monkeypox, emerging disease epidemiology, and travel medicine.

### 2.4. Ethical Consideration

This study was ethically reviewed and approved by the Research Ethics Committee at Security Forces Hospital Program in Holy Capital (HAP-02-K-052). The approval number is ECM#0488-220522. This study was conducted according to the Declaration of Helsinki. Participants were also assured of anonymity and the confidentiality of their responses to the questionnaire.

### 2.5. Data Analysis

The data were extracted, cleaned, coded, and analyzed using IBM SPSS version 21 (SPSS version 21.0; IBM Corporation, Armonk, NY, USA) statistical software. Statistical significance was set at a 0.05 level. Frequency distribution was performed for categorical variables expressed in numbers and percentages. This was an exploratory study. Therefore, we used the mean score of 14 as a cut-off point; a mean score of 14 or above was considered high, while less than 14 was considered low. Pearson’s Chi-squared test was used to compare response variables and explanatory variables. The associations between our study’s explanatory variables and knowledge about MPX as a dependent variable were assessed using a two-step binary logistic regression for good knowledge cut-off points. For each independent variable, the crude odds ratio (OR) and 95% confidence interval (CI) were initially computed. The adjusted analysis included all drivers or barriers that were significant in the unadjusted logistic regression analysis (defined with *p* 0.05), with the output being an adjusted odds ratio (aOR). The *p*-value was set at <0.05 for statistical significance.

## 3. Results

During the survey, 450 responses to the questionnaire were received, and 52 had to be excluded due to incomplete information or refusal to participate. The final analysis included 398 participants, with a response rate of 88.4%. Approximately 58% of the participants were under 35 years old, and 56.8% were male. More than half (51%) of the participants were single. Approximately 36.9% of respondents were residents, followed by general practitioners (28.9%), and regarding their medical specialties, 24.4% of participants were general medical, followed by other specialties (21.4%) and preventive medicine (17.8%). Most participants (84.2%) worked in the government sector. Approximately 39.4% of the study participants had medical experience ranging from 1–5 years. More than a third of participants (34.7%) worked in the western region of Saudi Arabia, followed by the central area (25.4%). Only 18.6% of the surveyed participants had ever received information about monkeypox in their medical education. There were 380 (95.5%) participants who had heard of HMPX before the survey, of whom 51% had received the information relatively recently, “within several days or weeks before the survey” (Table 1).

### 3.1. Physicians’ Knowledge about Monkeypox and Associated Determinants

We utilized the average of all scores. We assumed that a score of 14 or higher indicated good knowledge, while a score below 14 indicated poor knowledge. Only 219 (55%) out of 398 respondents had a good knowledge of monkeypox. Across some dimensions, most participants had an accurate knowledge of monkeypox. The majority of respondents’ answers (87.5%) said that monkeypox is not prevalent in Middle Eastern countries.

On the other hand, 75.2% of the participants indicated that monkeypox was not prevalent in Western and Central Africa. Almost all (94.7%) said a virus causes monkeypox, and more than 57.9% said the signs and symptoms of monkeypox and smallpox are not the same. However, other questions were answered incorrectly. Approximately 95% stated that a HMPX case had not been reported in Saudi Arabia. Besides symptomatic treatment, 50.6% of physicians mentioned that an antiviral is required in monkeypox management. Even though almost everyone who answered correctly said that monkeypox is caused by a virus, 15.3% said that an antibiotic is needed to treat monkeypox in humans (Table 2).

Table 3 shows the relationship between Saudi physicians’ sociodemographic characteristics and their knowledge of HMPX. The mean score’s standard deviation was 5.44 points. As a result, 14 was chosen as the cutoff, with a score of 14 or higher indicating good knowledge and a score of less than 14 indicating poor knowledge. A ‘good knowledge’ score for monkeypox was associated with an age under 35 years (*p* < 0.01), the female gender (*p* < 0.01), being a general practitioner (*p* = 0.04), working in the private sector (*p* < 0.01), and having information on HMPX during medical school or residency years (*p* = 0.01). An age under 30 was correlated with a knowledge score (60%). High knowledge scores of monkeypox were more common among female physicians (65.1%). Regarding the level of work, general practitioners (64.3%), residents (55.8%), consultants (47.2%), and lastly, (45.3%) specialists were correlated with higher scores, whilst working in the private sector was associated with higher scores when compared to government workers (74.6% and 51.3%). Physicians who learned about HMPX during medical school or residency had higher knowledge scores than others (67.6% vs. 52.2%).

The first logistic regression analysis showed that some demographic factors such as age, gender, marital status, level of work, and working in the private sector were associated with a good level of knowledge about HMPX (Table 4). Being a female physician and working in the private sector were associated with a good level of knowledge about HMPX (OR: 2.08; 95% CI: 1.38–3.12 and OR: 2.78; 95% CI: 1.52–5.10, respectively). Besides demographic factors, those physicians who had information on HMPX during medical school or residency education years were 1.91 times more likely to have a good level of knowledge about HMPX compared to those who had not (Table 4). Regarding, the factors associated with a poor level of knowledge about HMPX, they included physicians aged 36–56 years. Additionally, married physicians were less knowledgeable about HMPX compared to singles (OR: 0.09; 95% CI: 0.71–0.48).

All factors that were significant in the unadjusted model were included in the next logistic regression model. In the adjusted logistic regression model, only some factors were significantly associated with a good level of knowledge about HMPX. These included gender, marital status, working in the private sector, and having previous information on HMPX during medical school or residency (Table 4). Female physicians had three times higher the odds of a good level of knowledge about HMPX compared to male physicians. Moreover, physicians who worked in the private sector had almost four times the odds of having a good level of knowledge about HMPX compared to those who work in the governmental sector (OR: 3.76; 95% CI: 1.67–8.44).

The physicians who had previous information on HMPX during medical school or residency had almost a three times higher odds ratio of having a good level of knowledge about HMPX compared to those who did not (Table 4).

### 3.2. Physicians’ Attitude toward the Monkeypox Virus

The attitudes of physicians towards monkeypox are presented in Table 5. More than half of the participants (56%) agreed that monkeypox could add a new burden to the healthcare system worldwide. Moreover, most physicians (78.6%) were confident that the Saudi MOH and local population could control the monkey pox locally. Nearly two-thirds of the study sample (64.6%) agreed that monkeypox could be easily transmitted to Saudi Arabia, that the mass media may influence its prevention, and that the world’s populations can control the monkeypox epidemic. Additionally, 44.5% of those polled said they had no negative feelings about monkeypox disease. More than half of the participants (59.3%) agreed that travelling to monkeypox epidemic countries would be risky. Half of the participants (49.7%) agreed that monkeypox has enough prevention and control measures. Most participants wanted to know more about monkeypox, the epidemiology of emerging diseases, and travel medicine (71.1%, 70.9%, and 71.9%, respectively).

## 4. Discussion

Monkeypox represents a new challenge for physicians worldwide from multiple points of view, especially including its diagnosis, management and prevention, especially in regions where still cases have not been reported. For example, in the Middle East, and especially close to Saudi Arabia, few countries have reported cases, including the United Arab Emirates (16), Qatar (5), Jordan (1), Iran (1), Bahrain (1), among others. Israel and Lebanon have also reported cases (262 and 24, respectively). There is an inherent bias in developed countries that diseases that are endemic or emerge within less developed areas are largely restricted to those regions and as such education is limited in the medical profession. Education and knowledge about diseases that are in other countries usually is neglected, as the risk of importation and that they become endemic is very low. Our survey for monkeypox and the current COVID-19 pandemic as zoonotic infections that have spread worldwide are stark counterexamples and support a more intense global understanding of this phenomenon [22,23,24].

Beyond this, Saudi Arabia is a highly relevant global destination for business, sports, tourism, and religious visits (Hajj and Omrah), among other aspects [25]. Therefore, throughout the course of the subsequent weeks, imported cases will be observed and confirmed, and the implementation of proper preventive measures is highly crucial, as well as proper diagnosis and management. For these reasons, attending physicians in this country need to be prepared regarding the clinical and epidemiological aspects of this emerging viral disease. Preparedness in different countries and regions is critical [26]. In this setting, the results of this study are concerning. A significant proportion of the assessed physicians is not clear regarding the endemicity of monkeypox, as well as about its transmission, clinical differences with smallpox, chickenpox, and influenza, as well as the clinical evolution (e.g., skin lesion evolution) and the main associated findings. Also, there is a large gap regarding the therapeutic management of disease and vaccination. Such poor knowledge is influenced by an increasing age (worse in physicians when getting older than 36 years), gender (worse in males), level of work (specialists and consultants), work sector (worse in governmental areas), and previous medical training (worse in those that have not received information on monkeypox during medical school or residency years); additionally, there were no significant differences according to the specialty, years of experience, region of the country, if one had heard before about the disease, and if it was the first time they heard about it.

This means that massive continuing medical education on monkeypox in Saudi Arabia is critical at this moment, as revealed from this study. As previously shown during other epidemics and pandemics, such as the Zika and COVID-19 outbreaks, respectively [27,28], training and knowledge assessment is highly important, especially when such studies demonstrated significant improvement in the related and specific knowledge [25,26].

Even more, as has been demonstrated in recent clinical reports [29,30], the presentation of monkeypox relatively differs from the expression that was previously reported in African endemic-countries [31]. Then, it is clear that physicians, with confirmed and without confirmed cases, need to be not just prepared but aware of such differences that include the importance of sexual contact and the high occurrence of monkeypox among men who have sex with men, including people living with HIV, among other risk factors [32]. Additionally, over the course of May–July 2022, the assessment of monkeypox virus as a sexual pathogen has been under careful assessment. Some reports have identified it in sexual fluids, but still, confirmation of its replication in cell cultures and transmission is needed [33,34]. These findings also have significant implications for prevention and control that should be known by attending physicians in Saudi Arabia, as well as elsewhere.

Finally, as this is a relatively new disease outside Africa, even with a lack of research prior to 2022 [35], research in different aspects is needed, including its treatment and prevention, which also include the use of vaccines against this virus. The development of national clinical guidelines is key, as occurred with COVID-19 [36]. Then, evidence-based guidelines for monkeypox should be developed, and these should be implemented and widely promoted among physicians in Saudi Arabia and other countries, in order to provide the best available clinical management, especially because although most monkeypox cases will evolve without complications, these may occur, and even fatal cases in Africa have been reported, reaching a case fatality rate up to 10%. Early diagnosis, identification of risk factors, and prompt management of monkeypox are critical in reducing the risk of complicated cases and fatal outcomes.

Our research is groundbreaking on this hot topic of public health in Saudi Arabia. Our findings could have potential significance for Saudi health authorities. However, some limitations were unavoidable. Our study was self-reported and therefore subject to information bias, including a convenience sample and not a probabilistic one. Moreover, it was conducted through specific online platforms, which means we may not have reached other groups of the targeted population, such as other HCWs or physicians who do not use these platforms. Nationwide survey studies could be appropriate for digging out other factors related to this subject. Additionally, qualitative studies should be conducted to get a better, broader and more integrative understanding regarding the knowledge, attitude, and perceptions of healthcare workers in Saudi Arabia in association with monkeypox.

Finally, future research should concentrate on continuous education, raising awareness through programs, and developing strategies to effectively overcome identified factors contributing to the knowledge and attitudes about the monkeypox pandemic in Saudi Arabia by listening to physicians’ concerns and incorporating public and health perspectives in the planning of these policies and programs. 

## 5. Conclusions

A significant proportion of the Saudi physicians polled were unaware of monkeypox’s endemicity, transmission, and clinical differences from smallpox, as well as its clinical evolution and the most common associated findings. Furthermore, there is a significant knowledge gap between monkeypox therapeutic management and vaccination. The knowledge and attitudes of Saudi Arabian physicians about infection with monkeypox are poor and are impacted by a number of different circumstances. Training and more in-depth continuous education about this topic and any suspected emergent diseases are crucial. Evaluation of knowledge is vital, especially when studies reveal a considerable growth in related and specialized information, as was demonstrated by earlier epidemics and pandemics, such as Zika and COVID-19.

## Figures and Tables

**Table 1 vaccines-10-02099-t001:** Sociodemographic characteristics of physicians who participated in the study (*n* = 398).

Variable	Total	
	*n*	(%)
Age (years)		
26–35	230	57.8
36–45	129	32.4
46–56	24	6
Above 56	15	3.8
Gender		
Male	226	56.8
Female	172	43.2
Marital status		
Single	203	51
Married	195	49
Level of work		
General practitioner	115	28.9
Resident	147	36.9
Specialist	64	16.1
Consultant	72	18.1
Medical specialty		
General medical	97	24.4
Family medicine	60	15.1
Pediatrics	33	8.3
Internal medicine	36	9
Emergency medicine	4	1
Preventive medicine	71	17.8
Dermatology	12	3
Other	85	21.4
Work sector		
Governmental	335	84.2
Private	63	15.8
Medical practice experience		
Less than one year	107	26.9
1–5 years	157	39.4
More than five years	134	33.7
Region of work in Saudi Arabia		
Southern region	58	14.6
Northern region	36	9
Central region	101	25.4
Western region	138	34.7
Eastern region	65	16.3
Information about HMPX during medical education		
No	324	81.4
Yes	74	18.6
Heard about HMPX before		
No	225	56.5
Yes	173	43.5
First time you heard information about monkeypox		
I did not hear about it	18	5
Within several days or weeks ago	203	50.5
Within the last month or later	177	44.5

**Table 2 vaccines-10-02099-t002:** Responses of physicians who participated in the study to the knowledge questions about the monkeypox virus (*n* = 398).

Knowledge Questions	Correct *n* (%)	Incorrect *n* (%)
Q1. Is monkeypox prevalent in middle eastern countries?	349 (87.5)	50 (12.5)
Q2. Is monkeypox prevalent in Western and Central Africa?	99 (24.8)	300 (75.2)
Q3. Are there many human monkeypox cases in Saudi Arabia?	379 (95)	20 (5)
Q4. Is monkeypox a viral disease infection?	378 (94.7)	21 (5.3)
Q5. Is monkeypox a bacterial disease infection?	374 (93.7)	25 (6.3)
Q6. Is monkeypox easily transmitted human-to-human?	162 (40.6)	237 (59.4)
Q7. Could monkeypox be transmitted through a bite of an infected monkey?	193 (48.4)	206 (51.6)
Q8. Are travelers from America and Europe the primary source of imported cases of monkeypox?	167 (41.9)	232 (58.1)
Q9. Do monkeypox and smallpox have similar signs and symptoms?	168 (42.1)	231 (57.9)
Q10. Do monkeypox and chickenpox have similar signs and symptoms?	169 (42.4)	230 (57.6)
Q11. Is a flu-like syndrome one of the early signs or symptoms of human monkeypox?	79 (19.8)	320 (80.2)
Q12. Are rashes on the skin one of the signs or symptoms of human monkeypox?	37 (9.3)	362 (90.7)
Q13. Are papules on the skin one of the signs or symptoms of human monkeypox?	74 (18.5)	325 (81.5)
Q14. Are vesicles on the skin one of the signs or symptoms of human monkeypox?	117 (29.3)	282 (70.7)
Q15. Are pustules on the skin one of the signs or symptoms of human monkeypox?	157 (39.3)	242 (60.7)
Q16. Is diarrhea one of the signs or symptoms of human monkeypox?	215 (53.9)	184 (46.1)
Q17. Is lymphadenopathy (swollen lymph nodes) one clinical sign or symptom that could be used to differentiate between monkeypox and smallpox cases?	84 (21.1)	315 (78.9)
Q18. One management option for symptomatic monkeypox patients is to use paracetamol?	52 (13)	347 (87)
Q19. Are antivirals required in the management of human monkeypox patients?	197 (49.4)	202 (50.6)
Q20. Are antibiotics required in the management of human monkeypox patients?	61 (15.3)	338 (84.7)
Q21. Are people who got the chickenpox vaccine immunized against monkeypox?	264 (66.2)	135 (33.8)
Q22. Is there a specific vaccine for monkeypox?	278 (69.8)	120 (30.2)
Q23. Is there a specific treatment for monkeypox?	65 (16.4)	332 (83.6)

**Table 3 vaccines-10-02099-t003:** Association between sociodemographic variables and knowledge score among Saudi physicians who participated in the study (*n* = 398).

Sociodemographic Variables	Good Knowledge *n* (%) 219 (55)	Poor Knowledge *n* (%) 179 (45)	χ^2^	*p*
Age (years) 26–35 36–45	124 (54) 83 (64.3)	106 (46) 46 (35.7)	16.90	<0.01 *
46–56	10 (41.7)	14 (58.3)		
Above 56	2 (13.3)	13 (86.7)		
Gender Male Female	107 (47.3) 112 (65.1)	119 (52.7) 60 (34.9)	12.46	<0.01 *
Marital status Single Married	120 (59.1) 99 (50.2)	83 (40.9) 96 (49.8)	2.79	0.03 *
Level of work General practitioner Resident Specialist Consultant	74 (64.3) 82 (55.8) 29 (45.3) 34 (47.2)	41 (35.7) 65 (44.2) 35 (54.7) 38 (52.8)	8.28	0.04 *
Medical specialty General medical Family medicine Pediatrics Internal medicine Emergency medicine Preventive medicine Dermatology Other	60 (61.9) 33 (55) 12 (36.4) 23 (63.9) 2 (50) 36 (50.7) 6 (50) 47 (55.3)	37 (38.1) 27 (45) 21 (63.6) 13 (36.1) 2 (50) 35 (49.3) 6 (50) 38 (44.7)	8.32	0.30
Work sector Governmental Private	172 (51.3) 47 (74.6)	136 (48.7) 16 (25.4)	7.61	<0.01 *
Medical practice experience Less than 1 year 1–5 years More than 5 years	67 (62.6) 83 (52.9) 69 (51.5)	40 (37.4) 74 (47.1) 65 (48.5)	3.46	0.18
Region of work in Saudi Arabia Southern region Northern region Central region Western region Eastern region	35 (60) 14 (38.9) 53 (52.5) 76 (55.1) 41 (63.1)	23 (40) 22 (61.1) 48 (47.5) 62 (44.9) 24 (63.9)	6.42	0.17
Information on HMPX during medical school or residency years education No Yes	169 (52.2) 50 (67.6)	155 (47.8) 24 (32.4)	5.78	0.01 *
Heard about HMPX before No Yes	106 (47) 113 (65.3)	119 (53) 60 (43.7)	13.1	<0.01 *
First time you heard information about monkeypox I did not hear about it Within several days or weeks ago Within the last month or later	8 (44) 118 (58) 93 (52.5)	10 (56) 85 (42) 84 (47.5)	2.04	0.36

χ^2^ = Chi-squared test; *p* = *p*-value; * Statistically significant.

**Table 4 vaccines-10-02099-t004:** Multivariable linear regression analysis for estimates of predictors associated with getting a good knowledge score about MPOXV among physicians who participated in the study (*n* = 398).

			Good Knowledge		Univariate				Multivariate			
Item	*n*	(%)	*n*	(%)	*p*-Value	OR	95%CI		*p*-Value	OR	95%CI	
Age (years)												
26–35 (ref)	230	(57.8)	124	(54)		1				1		
36–45	129	(32.4)	83	(64.3)	0.02 *	0.46	0.24	0.89	0.46	0.71	0.28	1.77
46–56	24	(6)	10	(41.7)	<0.01 *	0.13	0.03	0.54	0.02 *	0.09	0.01	0.69
>56	15	(3.8)	2	(13.3)	0.07	0.33	0.10	1.09	0.17	0.27	0.04	1.77
Gender												
Male (ref)	226	(56.8)	107	(47.3)			1			1		
Female	172	(43.2)	112	(65.1)	<0.01 *	2.08	1.38	3.12	<0.01 *	3.69	2.13	6.39
Marital Status												
Single (ref)	203	(51)	120	(59.1)	1				1			
Married	195	(49)	99	(50.2)	<0.01 *	0.09	0.71	0.48	0.02 *	0.48	0.25	0.90
Level of work												
General practitioner (ref)	115	(28.9)	74	(64.3)	1				1			
Resident	147	(36.9)	82	(55.8)	0.16	0.69	0.42	1.15	0.83	0.87	0.25	3.01
Specialist	64	(16.1)	29	(45.3)	0.01 *	0.46	0.25	0.85	0.24	0.45	0.12	1.72
Consultant	72	(18.1)	34	(47.2)	0.02 *	0.49	0.27	0.90	0.82	1.20	0.24	6.14
Medical Specialty												
General Medical (ref)	97	(24.4)	60	(61.9)	1							
Family Medicine	60	(15.1)	33	(55)	0.37	0.75	0.39	1.45				
Pediatrics	33	(8.3)	12	(36.4)	0.06	0.35	0.15	0.79				
Internal Medicine	36	(9)	23	(63.9)	0.83	1.09	0.49	2.41				
Emergency Medicine	4	(1)	2	(50)	0.64	0.62	0.09	4.58				
Preventive Medicine	71	(17.8)	36	(50.7)	0.15	0.63	0.34	1.18				
Dermatology	12	(3)	6	(50)	0.43	0.62	0.19	2.05				
Other	85	(21.4)	47	(55.3)	0.37	0.76	0.42	1.38				
Work Sector												
Governmental (ref)	335	(84.2)	172	(51.3)	1				1			
Private	63	(15.8)	47	(74.6)	<0.01 *	2.78	1.52	5.10	<0.01 *	3.76	1.67	8.44
Medical practice experience												
Less than 1 year (ref)	107	(26.9)	67	(62.6)			1					
1–5 years	157	(39.4)	83	(52.9)	0.12	0.67	0.41	1.11				
More than 5 years	134	(33.7)	69	(51.5)	0.08	0.63	0.38	1.06				
Region of work in Saudi Arabia												
Southern region (ref)	58	(14.6)	35	(60)			1					
Northern region	36	(9)	14	(38.9)	0.40	0.42	0.18	0.98				
Central region	101	(25.4)	53	(52.5)	0.33	0.73	0.38	1.39				
Western region	138	(34.7)	76	(55.1)	0.49	0.81	0.43	1.50				
Eastern region	65	(16.3)	41	(63.1)	0.76	1.12	0.54	2.33				
Information on HMPX during medical school or residency years education												
No (ref)	324	(81.4)	169	(52.2)	1 1							
Yes	74	(18.6)	50	(67.6)	0.02 *	1.91	1.12	3.26	<0.01 *	2.81	1.44	5.46
Heard about HMPX before												
No (ref)	225	(56.6)	106	(47)	1							
Yes	173	(43.5)	113	(65.3)	0.11	1.38	0.93	2.07				
First time you heard information about monkeypox												
I did not hear about it (ref)	18	(5)	8	(44.4)	1							
Within several days or weeks ago	203	(50.5)	118	(58.1)	0.26	1.73	0.66	4.58				
Within last month or later	177	(44.5)	93	(52.5)	0.51	1.38	0.52	3.67				

OR = odds ratio. C.I. = confidence interval. * Statistically significant (*p* < 0.05). (ref) = reference.

**Table 5 vaccines-10-02099-t005:** Attitudes about monkeypox virus, emerging diseases and travel epidemiology among Saudi physicians who participated in the study (*n* = 398).

Sentence	Agree *n* (%)	No Opinion *n* (%)	Disagree *n* (%)
I am confident that the world’s population can control monkeypox worldwide	223 (56)	136 (34.2)	39 (9.8)
I am confident that the Saudi MOH and local population can control the monkeypox locally	313 (78.6)	66 (16.6)	19 (4.8)
I think that there are currently enough prevention and control measures for monkeypox	198 (49.7)	119 (29.9)	81 (20.4)
I have bad feelings toward the monkeypox virus that it might become a worldwide pandemic	100 (25.1)	121 (30.4)	177 (44.5)
I think that monkeypox can add a new burden on the healthcare system of the affected countries	190 (47.7)	131 (32.9)	77 (19.3)
I think monkeypox can be transmitted to Saudi Arabia	256 (64.6)	98 (24.6)	43 (10.8)
I think that mass media coverage of monkeypox may influence its worldwide prevention	244 (61.3)	131 (32.9)	23 (5.8)
I am interested in learning more about monkeypox	283 (71.1)	82 (20.6)	33 (8.3)
I am interested to learn more about the epidemiology of the new emerging diseases	282 (70.9)	85 (21.4)	31 (7.8)
I am interested in learning more about Travel Medicine	286 (71.9)	80 (20.1)	32 (8)
I think that it is dangerous to travel to the country’s epidemic with monkeypox	236 (59.3)	111 (27.9)	51 (12.8)

## Data Availability

The data presented in this study are available on reasonable request from the corresponding author.

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
