# Peer review of "Knowledge and Attitude Regarding Monkeypox Virus among Physicians in Saudi Arabia: A Cross-Sectional Study"

_vaccines, 2022, doi:10.3390/vaccines10122099_

Round 1

Reviewer 1 Report

Thank you for the manuscript. There are a number of issues I have raised in the attached pdf document but I wanted to congratulate your team on your effort and your findings 

Author Response

Responses to reviewers’ comments

Reviewer #1

Thank you for the manuscript. There are a number of issues I have raised in the attached pdf document but I wanted to congratulate your team on your effort and your findings 1. Respondent population- Lines 110-111- The population of Saudi medical providers was stated: “We determined the sample size based on recent data, estimating the total number of 110 physicians in Saudi Arabia to be 1047752.” I don’t believe this number is accurate based on my searching in medical demographics

Authors’ response: Thank you so much for this explanatory comment, we have concluded this number as per the finding of this statics organization. https://www.statista.com/statistics/629028/saudi-arabia-number-of-physicians-by-nationality. They wrote that number of Saudi and non-Saudi physicians is around 14000 in 2020. However, we modified these statements as requested by you.

  1. Survey. Consent- lines 121-122; Its stated consent was provided by completing the survey. This does not constitute consent for a survey study. Possible respondents should have been informed and consented PRIOR to attempting the survey. Usually, we provide a study information page prior to the questionnaire concerning the objective of the study, protections for identity, how the responses are collected and stored and assurance that the survey is voluntary and has no effect on their employment status. The respondent then acknowledges these points and provides their consent then is allowed to proceed with the actual survey

Authors’ response: We have addressed this on page 3 lines 120-126, the last paragraph of the "Data collection process" of the method has been updated to indicate the detailed process of the informed consent process. In summary " Each respondent was requested for their informed consent by clicking the consent statement before attempting to fill out the responses. The informed consent statement given to the respondents said, "I hereby, after reading the aims of the study, engage in the survey supplying my information by addressing questions intelligently and voluntarily." They completed the survey and clicked the "submit" button to send it to our platform for gathering data. To submit a legitimate response, all of the questions have to be answered".

  1. b. Validity- lines 124-129: as this was a team developed survey additional validation of the survey a to internal and external validity should be provided included with a Cronbach’s alpha or other documented measure.

Authors’ response: Thank you, and we have added further details requested by reviewers.

  1. c. Questionnaire responses- Line 137-142: The use of the word NATURAL in the survey repone is not typical of use as a survey response and should be clarified- “neither agree or disagree”. The 3 response choices of the Likert scale are somewhat limiting as they do not reflect a more granular sense of the understanding of the respondent population. Scales are usually 5 in number form such as “strongly agree, agree, unsure, disagree, and strongly disagree” as an example. How do the authors provide support for a three-answer choice?.

Authors’ response: Thank you; we agree with your opinion. However, we have modified our survey on this particular section based on a previously published article in Saudi Arabia, which was done by Khalil et al. Journal of Infection and Public Health, January 1st, 2018;11(1):18-23, to preserve the understood process of this emerging monkeypox. That study was done on Zika during its outbreak in 2014. However, we corrected our typo and changed "natural" to "No opinion"

  1. d. Nonspecific statement- “bad feelings” Line 138-141: “These statements included their opinions about the ability of the world's populations to control the monkeypox epidemic, the presence of suitable preventive and control measures, and whether they had bad feelings towards monkeypox.” I cannot believe any physician would have any “good feelings “about MPox and so this statement does not convey any precise determinate and how the physician classifies their actual opinion of MPox. Redefining this metric should be done.

Authors’ response: Thank you for your advice. we have modified it as per request.

  1. data analysis and results Scoring lines 154-155: “We used the mean score of 14 as a cut point; a mean score of 14 or above was considered high, while less than 14 was considered low.” The scoring categories are very narrow and perhaps fail to capture on a more granular level the spread of knowledge within the respondents- Recalculation based on additional categories of scores would provide a better idea of the actual knowledge level within the respondents

Authors’ response: Thank you for this note. We conducted our research as early in the monkeypox outbreak as possible, by using the model study that others have conducted on this topic as a guide. Therefore, for statistical analysis, the levels of knowledge were dichotomized into good and poor based on two modified Bloom’s cutoff points. The difference between their work and our study was based on several questions that have been modified after the piloting process. Kindly, find their justification and reference can be found below. 

" a 70% and 80% of the total score (i.e. if a participant answered correctly 15 and 17 out of the total 21 questions, respectively)"

Harapan H, Setiawan AM, Yufika A, Anwar S, Wahyuni S, Asrizal FW, Sufri MR, Putra RP, Wijayanti NP, Salwiyadi S, Maulana R, Khusna A, Nusrina I, Shidiq M, Fitriani D, Muharrir M, Husna CA, Yusri F, Maulana R, Andalas M, Wagner AL, Mudatsir M. Knowledge of human monkeypox viral infection among general practitioners: a cross-sectional study in Indonesia. Pathog Glob Health. 2020 Mar;114(2):68-75. doi: 10.1080/20477724.2020.1743037. Epub 2020 Mar 23. PMID: 32202967; PMCID: PMC7170312

  1. Effect size- lines 156-157: “Pearson's Chi-square test was used to compare response variables and explanatory variables. P-value was set at t <0.05 for statistical significance.”Reliance on P values alone has been criticized as inadequate and does not reflect clinically relevant differences such as effect size. The authors should provide some statement as to what they believe is relevant effect size. Results- concern for sampling- lines 161-163

Authors’ response: Thank you. We used the chi-square test and the P value to determine the correlation between the variables. However, we also used frequency distributions for categorical variables expressed in numbers and percentages. As suggested by the reviewer, we run a logistic regression analysis.

8.“Approximately 57% of the 161 participants were under 30 years old, and 56.8% were male. More than half (51%) of the 162 participants were single. Approximately 36.9% of respondents were residents” As mentioned previously the respondents’ demographics raise concern as to the age and clinician status. The authors need to provide a Saudi Physician demographic so the reader can place the results in context.

Authors’ response: Thank you for your advice. we have modified it as per request.

9.Table 2- The designation of the categorical outcomes as “correct versus incorrect” does not provide a more nuanced outcome that would come from designation using additional categories of the scoring. Additionally, it would have been insightful to determine the strength of their responses. Statements are now being incorporated in surveys to determine this important issue which has been translated from medical decision making. Examples are – “how confidant are you that your knowledge of MPox symptoms is accurate” etc. It provides an additional layer of understanding of the respondents in the survey . The questions should have also asked about the knowledge of spread between gay and bisexual men as this was the early and continued risk determinant

Authors’ response: Thank you for this suggestion. Again, we conducted this paper in the early stage of this monkeypox outbreak. Our survey was based on available data, information, and a validated survey at that time. All your suggestions about this point were done in another study about this topic among all Saudi healthcare workers and the paper is under review, for sure your suggestion will be highly appreciated and will be considered in that study.

  1. Data is presented as the wo categories < 30 and > 30. As mentioned earlier that designation does not adequately demonstrate the actual age range for the responses. Suggest- < 30, 30- 50, 50- 60, >60 if possible. Determining where respondents obtain their education is critical but not determined. As the authors will be targeting certain physician populations for health education interventions it would be important. For example, in the US we improve resident knowledge by requiring resident program education whereas educating a 55-year-old Saudi physician would depend on what source they use for understanding new and emerging health issues

Authors’ response: Thank you, we have reanalyzed our data and improved this as per reviewers' suggestions. for this suggestion.

11.Table 4- This is perhaps the most insightful and interesting part of the survey. I would as previously stated change the term “natural” to “unsure ” if accurate?

Authors’ response: Thank you, we have corrected it and modified it accordingly.

  1. Discussion- lines 228-230: Th following statement does not have scientific rigor and should be revised. “The Spread of emerging diseases, including zoonoses and potentially new sexually transmitted infections, such as monkeypox, seems a never-ending story, especially after the COVID-19 pandemic” There is an important sentiment in this statement and a suggested revision is- “There is an inherent bias in developed countries that disease that are endemic or emerge within less developed areas are largely restricted to those regions and as such education is limited in our medical profession. Our survey for MPx and the current CPVID-19 pandemic as zoonotic infections that have spread worldwide are stark counter examples and support more intense global understanding of this phenomenon.” I would also not state MPox as a sexually transmitted disease per se as it may limit the practitioner’s appreciation and relevance of he condition.

Authors’ response: We thank the reviewer for this comment, we agree with his point of view. We have corrected this statement and it and modify it according to your suggestion.

  1. Line 232-233: “Then, imported cases will be eventually observed and confirmed over the course of the next following weeks. I don’t understand the relevance of this statement

Author's response: We have updated this statement in a more meaningful way.

  1. Lines 241-246: The stated conclusion need in this paragraph needs to be modified based on my earlier comments and suggestions

Authors’ response: we have modified it per your suggestions.

Reviewer 2 Report

The authors present a study on healthcare provider knowledge and attitudes in Saudi Arabia.

I would like to share the following comments with the authors:

The introduction should be rewritten. The authors provide monkeypox information rather lengthy, this can be shorted and they should refer to existing summaries. Lines 44-95 can be shortened to one paragraph and referenced to a high level publication, such as Guarner et al 2022 in JAMA. What is missing, on the contrary is a proper review of the existing literature on MPX knowledge and attitudes among healthcare providers from other countries, e.g. Sallam et al. 2022 in Healthcare, or Ajman et al, 2022 in Vaccines, or Bass et al 2013 on MPX detection training. Also, a research question is missing and should be spelled out better.

The provided chi2 tests are not the accurate tests for contingency tables (table 4) with more than 4 cells, the authors should run a logistic regression analysis. Also the multitude of single tests leads to an overestimation of significant findings.

The General Discussion needs to be revised based on a proper analysis. Also, the GD should start off with a summary of the main findings and only then embed these findings in the study contexts, discussion of the findings in relation to other studies (see my point above) needs to be included. The narrative about Saudi Arabia should be placed under Conclusion.

Minor points:

The authors give a number of more than 1 million physicians in Saudi Arabia (line 111) without a reference. I find that highly uncredible given a number 38mil inhabitants. My quick search resulted in app. 114k physicians. Please provide a correct number.

The General Discussion should contain a Limitation section. Here for example, the authors shoudl discuss the self-selection bias of their sampling method

Author Response

Responses to reviewers’ comments

Reviewer #2

Introduction

The authors present a study on healthcare provider knowledge and attitudes in Saudi Arabia. I would like to share the following comments with the authors:

The introduction should be rewritten. The authors provide monkeypox information rather lengthy, this can be shorted and they should refer to existing summaries. Lines 44-95 can be shortened to one paragraph and referenced to a high level publication, such as Guarner et al 2022 in JAMA. What is missing, on the contrary is a proper review of the existing literature on MPX knowledge and attitudes among healthcare providers from other countries, e.g. Sallam et al. 2022 in Healthcare, or Ajman et al, 2022 in Vaccines, or Bass et al 2013 on MPX detection training. Also, a research question is missing and should be spelled out better.

Authors’ response: We have updated the introduction and added a paragraph to clearly indicate the importance of our study as well as a clear statement about the aim of the study. Moreover, we have included several most up-to-date references relating to the study’s theoretical implications and expounded the research gaps.

The provided chi2 tests are not accurate tests for contingency tables (table 4) with more than 4 cells, the authors should run a logistic regression analysis. Also, the multitude of single tests leads to an overestimation of significant findings.

Authors’ response: Thank you so much. We did not run chi2 tests in table 4. We just measure the attitudes of physicians towards monkeypox in Table 4 by number and percentage according to their answers. However, we run a logistic regression analysis to compare the significant factors associated with knowledge scores.

The General Discussion needs to be revised based on a proper analysis. Also, the GD should start off with a summary of the main findings and only then embed these findings in the study contexts, discussion of the findings in relation to other studies (see my point above) needs to be included. The narrative about Saudi Arabia should be placed under Conclusion.

Authors’ response: Thank you for your advice. we have modified it as per request

Minor points:

The authors give a number of more than 1 million physicians in Saudi Arabia (line 111) without a reference. I find that highly uncredible given a number 38mil inhabitants. My quick search resulted in app. 114k physicians. Please provide a correct number.

Authors’ response: Thank you so much for this explanatory comment, we have corrected it. Our updated number was based on the findings of this statics organization. https://www.statista.com/statistics/629028/saudi-arabia-number-of-physicians-by-nationality. They wrote that number of Saudi and non-Saudi physicians is around 14000 in 2020. However, we modified these statements as requested by you.

The General Discussion should contain a Limitation section. Here for example, the authors shoudl discuss the self-selection bias of their sampling method

Authors’ response: We thank the reviewer for this comment. We have updated the discussion. Now limitation paragraph 

Round 2

Reviewer 1 Report

Excellent revisions 

Author Response

Responses to reviewers’ comments

Reviewer #1

Excellent revisions

Authors’ response: Thank you so much for this compliment's expression.

Reviewer 2 Report

I have seen a previous version of this manuscript and I think that the MS has been improved. I would like to share the following comments with the authors:

The regression analysis presented does not follow the usual analysis steps (or at least is not communciating them properly). Now it seems that the authors ran a multivariable regression only, given that they report aOR. The authors need to conduct a univarible analysis first for all predictors and then indicate what is their cut off criterion for inclusion in a multivariable model. Both need to be presented, usually in two columns next to each other. The General Discussion needs to be adjusted according to the results.

The authors have now included a Limitation section which is good, but the presentation is off. Of course the authors can also highlight the benefits of their work, but I would highly recommend to tone that section down. This is research is neither rocket science, nor the world formula, it is merely a survey among healthcare professionals in one country on a one disease. Also the Limitations read a bit generic, and should be more to the point. What is the sampling bias precisely and what are potential consequences. Also: the proposed qualitative studies can never be representative, unless the authors seek to interview hundreds of HCW. This needs to be adjusted.

Minor point: the abbreviation used for monkeypox should be consistent, now it is HMPX, but also MPOXV (Table 5)

Author Response

Responses to reviewers’ comments

Reviewer #2

Introduction

I have seen a previous version of this manuscript and I think that the MS has been improved. I would like to share the following comments with the authors:

The regression analysis presented does not follow the usual analysis steps (or at least is not communciating them properly). Now it seems that the authors ran a multivariable regression only, given that they report aOR. The authors need to conduct a univarible analysis first for all predictors and then indicate what is their cut-off criterion for inclusion in a multivariable model. Both need to be presented, usually in two columns next to each other. The General Discussion needs to be adjusted according to the results.

Authors’ response: Thank you for your suggestion. we have modified it as per request.

The authors have now included a Limitation section which is good, but the presentation is off. Of course the authors can also highlight the benefits of their work, but I would highly recommend to tone that section down. This is research is neither rocket science, nor the world formula, it is merely a survey among healthcare professionals in one country on a one disease. Also the Limitations read a bit generic, and should be more to the point. What is the sampling bias precisely and what are potential consequences. Also: the proposed qualitative studies can never be representative, unless the authors seek to interview hundreds of HCW. This needs to be adjusted.

Authors’ response: We have edited the Discussion, and discuss more in detail certain aspects regarding the relative importance of this study in context. This survey is key in identifying a significant gap that potentially may affect the way in which imported cases and further epidemics of MPX in the country are dealt with.

Minor point: the abbreviation used for monkeypox should be consistent, now it is HMPX, but also MPOXV (Table 5).

Authors’ response: We thank the reviewer for this comment. We have updated the discussion.